# Safety and Immunogenicity of Enterovirus 71 Vaccine (Vero Cell) Administered Simultaneously with Trivalent Split-Virion Influenza Vaccine in Infants Aged 6–7 Months: A Phase 4, Randomized, Controlled Trial

**DOI:** 10.3390/vaccines11040862

**Published:** 2023-04-18

**Authors:** Yanhui Xiao, Xue Guo, Min Zhang, Yaping Chen, Yanyang Zhang, Xiaoqing Yu, Linyun Luo, Haiping Chen, Weichai Xu, Haibo Liu, Lixia Wu, Renwu Hou, Yong Ma, Lin Long, Jiewei Ruan, Wei Chen, Xiaoming Yang

**Affiliations:** 1Medical Affairs Department, China National Biotec Group Company Limited, No. 2, Shuangqiao Street, Chaoyang District, Beijing 100024, China; 2Medical Affairs Department, Changchun Institute of Biological Products Company Limited, Changchun 130012, China; 3Immunisation Programme Department, Zhejiang Provincial Center for Disease Control and Prevention, Hangzhou 310051, China; 4Institute for Communicable Disease Control and Prevention, Henan Provincial Center for Disease Control and Prevention, Zhengzhou 450016, China; 5Institute of Expanded Programme on Immunization, Guizhou Provincial Center for Disease Control and Prevention, Guiyang 550004, China; 6Immunisation Programme Department, Liandu District Center for Disease Control and Prevention, Lishui 323000, China; 7Immunisation Programme Department, Boai County Center for Disease Control and Prevention, Jiaozuo 454450, China; 8Immunisation Programme Department, Qianxinan Prefecture Center for Disease Control and Prevention, Qianxinan 562400, China; 9Medical Affairs Department, Wuhan Institute of Biological Products Company Limited, Wuhan 430070, China; 10National Engineering Technology Research Center for Combined Vaccines, Wuhan Institute of Biological Products Company Limited, Wuhan 430070, China

**Keywords:** enterovirus 71 vaccine (Vero cell), trivalent split-virion influenza vaccine, administered simultaneously, safety, immunogenicity

## Abstract

**Objective**: To assess the immunogenicity and safety of the enterovirus 71 vaccine (Vero cell) (EV71 vaccine) and trivalent split-virion influenza vaccine (IIV3). **Methods**: Healthy infants aged 6–7 months were recruited from Zhejiang Province, Henan Province, and Guizhou Province and randomly assigned to the simultaneous vaccination group, EV71 group, and IIV3 group at a ratio of 1:1:1. Then, 3 mL blood samples were collected before vaccination and 28 days after the second dose of vaccine. Cytopathic effect inhibition assay was used to detect EV71 neutralization antibody, and cytopathic effect inhibition assay was used to detect influenza virus antibody. **Results**: A total of 378 infants were enrolled and received the first dose of vaccine and were included in the safety analysis, and 350 infants were involved in the immunogenicity analysis. The adverse events rates were 31.75%, 28.57%, and 34.13% in the simultaneous vaccination group, EV71 group, and IIV3 group (*p* > 0.05), respectively. No vaccine-related serious adverse events were reported. After two doses of EV71 vaccine, the seroconversion rates of EV71 neutralizing antibody were 98.26% and 97.37% in the simultaneous vaccination group and the EV71 group, respectively. After two doses of IIV3, the simultaneous vaccination group and the IIV3 group, respectively, had seroconversion rates of 80.00% and 86.78% for H1N1 antibody, 99.13% and 98.35% for H3N2 antibody, and 76.52% and 80.99% for B antibody. There was no statistically significant difference in the seroconversion rates of influenza virus antibodies between groups (*p* > 0.05). **Conclusions**: The coadministration of EV71 vaccine and IIV3 has good safety and immunogenicity in infants aged 6–7 months.

## 1. Introduction

Enterovirus 71 (EV71) is the main cause of hand–foot–mouth (HFMD) disease in infants and children under 5 years old [1]. The severe and fatal cases of HFMD are mainly caused by EV71 infection [2]. Influenza is an acute respiratory infectious disease caused by influenza virus. Children under 5 years old have a high risk of severe illness after infection with influenza [3,4,5]. Influenza vaccine and EV71 vaccine are effective means to prevent influenza and hand, foot, and mouth disease caused by EV71, which can significantly reduce the risk of infection, severe disease, and death [6,7].

Children between the ages of 6 months and 3 years receive two doses of the trivalent-split-virion influenza vaccine (IIV3) with two doses given 2–4 weeks apart, which contains 7.5 μg of hemagglutinin for each influenza strain. The EV71 vaccine is approved for use in China at 6 months to 3 years of age, with 4 weeks apart. The safety and immunogenicity of EV71 vaccine and IIV3 was evaluated in the Zhejiang, Henan, and Guizhou provinces in 2019, considering that the applicable populations and immunization procedures of the two vaccines are the same.

## 2. Methods

### 2.1. Study Design and Participants

In the Zhejiang, Henan, and Guizhou provinces, 378 healthy infants aged 6–7 months were recruited and randomly assigned to the simultaneous vaccination group, EV71 group, and IIV3 group at a ratio of 1:1:1. Infants in the simultaneous vaccination group were given one dose of EV71 vaccine (0.5 mL per dose) and one dose of IIV3 (0.25 mL per dose) at the same time on day 0 and day 28. On the days 0 and 28, infants in the EV71 group received one dose (0.5 mL per dosage) of the EV71 vaccine. Infants in the IIV3 group received one dose of IIV3 (0.25 mL per dose) on day 0 and 28. Then, 3 mL of blood samples were taken from each infant on days 0 and 56. This study was approved by the Ethics Committee of the Centers for Disease Control and Prevention of Zhejiang Province, Henan Province, and Guizhou Province.

### 2.2. Inclusion Criteria and Exclusion Criteria

Inclusion criteria: 6–7 months old on the day of enrollment; informed consent was signed by the legal guardian; the time interval between the last vaccination ≥ 14 days; infants who had not received the EV71 vaccine and influenza vaccine of the current season and who have no previous history of EV71 disease; and the body temperature was ≤37.0 °C.

Exclusion criteria for the first dose: infants who were allergic to any component of the vaccine; infants who had a history of severe allergy reaction to any vaccine; infants who had a history of convulsion, epilepsy, encephalopathy, psychosis; infants who suffered from immunodeficiency, malignant tumor treatment, immunosuppressive therapy (oral steroids), and HIV-related low immunity; infants who were injected with nonspecific immunoglobulin within 3 months prior to enrollment; infants who had a history of thrombocytopenia or other coagulation disorders; and infants who suffered from infectious, suppurative, and allergic skin diseases.

Exclusion criteria for the second dose: infants with severe allergic reactions or serious adverse reactions after the first vaccination.

### 2.3. Vaccines

EV71 vaccine was manufactured by Wuhan Institute of Biological Products (lot number: 201810076), containing ≥3.0 EU of antigen with a seed virus of EV71 strain AHFY087VP5 (genotype C4). EV71 vaccine was prepared by inoculating Vero cells with enterovirus EV71, cell culture, harvest, concentration, purification, inactivation, and alumina adsorption. The EV71 vaccine was administered intramuscularly into the anterolateral thigh of the infant. Egg-based IIV3 was produced by Changchun Institute of Biological Products (lot number: T20190701), containing 7.5 hemagglutinin of per influenza virus strains. The hemagglutinins used in IIV3 were A/Brisbane/02/2018(H1N1) pdm09-like virus, A/Kansas/14/2017(H3N2)-like virus, and B/Colorado/06/2017-like virus (B/Victoria/2/87 lineage). The IIV3 was administered intramuscularly into the deltoid muscle of the upper arm of the infant.

### 2.4. Randomization and Masking

An independent statistician generated a random number using block randomization (block size:9) with SAS 9.4. An open-label trial was carried out to decrease needless injections. Infants, parents, and investigators were not masked for the study group assignment, but laboratory technicians were.

### 2.5. Safety Assessment

After vaccination, all participants should receive 30 min observation for adverse events (AEs). The guardian was given a diary card to keep track of any AEs that occurred within 28 days of the immunization. Symptoms of systemic solicited AEs include fever, rash, irritability, vomiting, diarrhea, drowsiness, etc. Symptoms of solicited AEs include pain, redness, swelling, induration, etc.

### 2.6. Immunogenicity Assessment

The cytopathic inhibition method was used to identify the neutralizing antibody titers of EV71 [8]. The serum samples were inactivated at 56 °C for 30 min, and then two-fold serially diluted beginning with 1:8. Then, 50 μL of a 100-TCID_50_ virus (AHFY087VP5 strain) suspension was added to each well of the 96-well plates containing 50 μL of serially diluted serum. After incubation at 37 °C for 2 h, 100 μL of rhabdomyosarcoma cell suspension at a final cell concentration 1 × 10^5^ mL were added to the mixture. The cultures were then incubated in CO^2^ incubators for 7 days at 35 °C. Cytopathic effect inhibition was observed under a light microscope. Neutralizing antibody titers were defined as the reciprocal of the highest dilution capable of inhibiting 50% of the cytopathic effect inhibition. Seropositive of EV71 neutralizing antibody was defined as neutralizing antibody titer ≥ 1:8. EV71 seroconversion was defined as neutralizing antibody titer < 1:8 before vaccination and neutralizing antibody titer ≥ 1:8 after vaccination or the titer of neutralizing antibody before vaccination was ≥1:8, and the titer of neutralizing antibody increases by ≥4 times after vaccination. 

Influenza virus antibody titers were detected using hemagglutination inhibition (HI) tests [9]. The protocol of HI testing to measure antibody responses is briefly described. The serum samples were melted at room temperature, then cholera filtrate was added, incubated in a 37 °C incubator for 18 h, and inactivated in a 56 °C water bath for 50 min. After the serum sample temperature reached room temperature, 50% chicken red blood cells were added, mixed thoroughly, and kept at 4 °C for the night. Serum samples were diluted 1:5 and diluted 2-fold 11 times serially. Then, 4 units of hemagglutinin antigen [A/Brisbane/02/2018(H1N1) pdm09-like virus, A/Kansas/14/2017(H3N2)-like virus, and B/Colorado/06/2017-like virus (B/Victoria/2/87 lineage)] were added, vortexed, and mixed, then kept for 45 min. Then, 1% chicken red blood cells were added and mixed, and kept for 45 min. The highest serum dilution of completely inhibiting erythrocyte agglutination was used as the HI titer. Seropositivity for influenza virus antibody before immunization was defined as antibody titer ≥ 1:10. Seroprotection for influenza virus antibody after immunization was defined as antibody titer ≥ 1:40 [10]. The seroconversion of influenza virus antibody was defined as the antibody titer before vaccination < 1:10 and the antibody titer after vaccination ≥ 1:40 or the antibody titer before vaccination was ≥1:10, and the antibody titer after vaccination increased by ≥4 times. The China National Institute for Food and Drug Control assessed the neutralizing antibodies to EV71 and the hemagglutination inhibition antibodies to the influenza virus.

### 2.7. Statistical Analysis

Statistical analysis was performed using STATA15 software. Participants who were enrolled and completed vaccination and blood collection as protocol design were included in the immunogenicity data set (Per Protocol Set, PPS). Participants who received at least one dose vaccine and completed at least one adverse event collection were included in the safety analysis. The antibody titer was descripted by geometric mean titer (GMT) and 95% confidence interval (95% CI). The GMTs were compared between groups using the T-test or a nonparametric test. An χ2 test or Fisher’s exact test was used for comparison of rates between groups. *p* < 0.05 was statistically significant.

## 3. Results

### 3.1. Study Participants

Between September 2019 and June 2020, 126 infants were enrolled in the simultaneous vaccination group, 126 infants were enrolled in the EV71 group, and 126 infants were enrolled in the IIV3 group. A total of 378 infants received the first dose of vaccine and were included in the safety analysis. Finally, 350 infants were involved in the per-protocol analysis (Figure 1).

The average age of participants in the simultaneous vaccination group, EV71 group, and IIV3 group was 6.56, 6.51, and 6.64 months old, and the male proportion was 50.43%, 44.74%, and 44.63%, respectively.

### 3.2. Safety Results

The incidence of AE rates within 28 days were 31.75%, 28.57%, and 34.13% in the simultaneous vaccination group, EV71 group, and IIV3 group, respectively. The systemic AE rates were 30.95%, 27.78%, and 32.54%, and the local AE rates were 1.59%, 0.79%, and 3.17%, respectively. In the three groups, there was no statistically significant difference in the rate of AEs (*p* > 0.05) (Table 1). Fever was the most common AE symptom. The fever rates were 16.67%, 15.87%, and 12.70% in the simultaneous vaccination group, EV71 group, and IIV3 group (*p* = 0.648). In terms of fever severity, 90.7% of participants with fever had a maximum body temperature of <39 °C. No serious adverse events related to vaccination were reported during the study. 

### 3.3. EV71 Antibody Results

Before vaccination, the seropositive rates for the EV71 neutralizing antibody were 33.04% [38/115] and 35.09% [40/114], and GMTs were 6.04 and 6.40 in the simultaneous vaccination group and EV71 group. There was no significant difference between the groups (*p* > 0.05).

After two doses of EV71 vaccine, both the simultaneous vaccination group and the EV71 group had 100% seropositivity for the EV71 neutralizing antibody, seroconversion rates of 98.26% [113/115] and 97.37% [111/114], and GMTs of 414.25 and 514.84, respectively. No difference between the two groups could be considered statistically significant (*p* > 0.05) (Table 2).

### 3.4. Influenza Virus Antibody Results

Before vaccination, for the simultaneous vaccination group and IIV3 group, the seropositive rates of H1N1 antibody were 5.22% [6/115] and 3.31% [4/121], the GMTs of H1N1 antibody were 5.31 and 5.17; the seropositive rates of H3N2 antibody were 4.35% [5/115] and 9.92% [12/121], the GMTs of H3N2 antibody were 5.18 and 5.39; and the seropositive rates of B antibody were 1.74% [2/115] and 2.48% [3/121], the GMTs of B antibody were 5.06 and 5.09. There was no significant difference between the groups (*p* > 0.05).

After two doses of IIV3, in the simultaneous vaccination group and IIV3 group, the seroprotection rates of H1N1 antibody were 80.87% [93/115] and 86.78% [105/121], the seroconversion rates of H1N1 antibody were 80.00% [92/115] and 86.78% [105/121], and the GMTs were 79.52 and 88.69, respectively. The seroprotection and seroconversion rates of H3N2 antibody in the simultaneous vaccination group and IIV3 group were 99.13% [114/115] and 98.35% [119/121], and the GMTs were 142.69 and 125.79, respectively. The seroprotection and seroconversion rates of B antibody in the simultaneous vaccination group and IIV3 group were 76.52% [88/115] and 80.99% [98/121], and the GMTs were 49.10 and 48.88, respectively. There was no significant difference between the groups (*p* > 0.05) (Table 3).

## 4. Discussion

EV71 vaccine has been widely used in children aged 6 months to 3 years old in China. Several post-marketing studies have shown that the EV71 vaccine has good immunogenicity, safety, efficacy, and antibody persistence. The results of EV71 vaccine effectiveness after 2 years revealed that the overall effectiveness to prevent HFMD during the 2-year follow-up period was 94.84%, and the GMT of participants remained high in two years [11]. A 5-year persistence study of EV71 vaccination showed that the GMTs of EV71 neutralizing antibody of the vaccine group was significantly higher than the placebo-controlled group (369.57 vs. 55.58, *p* < 0.01), and EV71 antibodies still maintained good immune persistence 5 years after two doses of immunization [12]. A large-scale safety monitoring study in Zhejiang Province found that the reported rate of AEFI was 267.90 per 100,000 persons, and there were no serious AEFI cases, and most common reactions were fever, redness, and induration, indicating that EV71 vaccine was safe [13]. A real-world effectiveness study showed that the average morbidity rate of EV71-related HFMD in Chengdu (Sichuan Province) in 2017–2018 was 60% lower than that in 2011–2017 when the EV71 vaccine was not available, and the number of severe HFMD cases was 52% lower, demonstrating that EV71-related HFMD significantly decreased after vaccination [14].

Previous studies have shown that when the EV71 vaccine was administered simultaneously with recombinant hepatitis B vaccine, group A meningitis polysaccharide vaccine, measles rubella vaccine, and Japanese encephalitis vaccine, the seroconversion rates ranged from 94.6%~99.6%; seroconversion rate and GMT of EV71 neutralizing antibody were not interfered with and did not increase the safety risk [15,16,17,18]. A single-center randomized controlled trial on the immunogenicity and safety of the simultaneous administration of the EV71 vaccine (dose 1) with recombinant hepatitis B vaccine on day 1 and EV71 vaccine (dose 2) with group A meningococcal polysaccharide vaccine on day 30 showed that the seroconversion rates of EV71 neutralizing antibody for the simultaneous vaccination group and EV71 group were 98.56% and 98.61% (*p* = 1.000) [15]. A single-center randomized controlled trial on the immunogenicity and safety of the EV71 vaccine coadministered with measles–mumps–rubella vaccine and live-attenuated Japanese encephalitis vaccine found that the seroconversion against EV71 found in the coadministration group and EV71 vaccination alone group were 97.27% and 97.32% (*p* = 1.000) [16]. A multi-center randomized controlled trial that was conducted in China showed that the seroconversion rates of antibodies against EV71 were 98.44%, 94.57%, 99.47%, 98.45%, and 97.93% in EV71 vaccine and hepatitis B virus vaccine simultaneous administration group, EV71 vaccine and group A meningococcal polysaccharide vaccine simultaneous administration group, EV71 vaccine and measles–rubella combined vaccine simultaneous administration group, EV71 vaccine and Japanese encephalitis vaccine simultaneous administration group, and the EV71 vaccine separate administration group, respectively [17]. A trial of IIV3 (0.25 mL dosage) conducted in Hubei showed that seropositive rates of H1N1, H3N2, and B were 73.3%, 86.3%, and 65.8% [19], which were lower than our results. Previous studies on the simultaneous vaccination of influenza vaccine mainly focused on the studies of the influenza vaccine and the 23-valent pneumococcal polysaccharide vaccine, and the results showed that the coadministration of the influenza vaccine and the 23-valent pneumococcal polysaccharide vaccine has good immunogenicity and safety in children. [20].

Our study evaluated the immunogenicity and safety of the EV71 vaccine simultaneously administered with IIV3 in infants aged 6–7 months. The seroconversion rate of EV71 neutralizing antibody was above 95%, the rates of influenza virus H1N1, H3N2, and B antibody seroconversion were greater than 80%, 98%, and 75%, respectively, and the rates of positive rotation and adverse events were comparable between the groups. Therefore, the simultaneous vaccination of the EV71 vaccine and IIV3 in infants aged 6 to 7 months was immunogenic and safe.

Our study has some limitations. Our study did not evaluate the immunogenicity and safety of the EV71 vaccine administered simultaneously with quadrivalent influenza vaccine, and future studies need to be conducted. Due to limitations in the content of safety data collection, we did not report the duration of adverse events.

In China, the applicable population and immunization procedures for the EV71 vaccine and IIV3 were completely matched. The coadministration of the EV71 vaccine and IIV3 in infants aged 6–7 months with a busy vaccination schedule could increase vaccine availability and reduce the risk of influenza and EV71 infection. Our findings showed that the EV71 vaccine and the IIV3 vaccine can be administered simultaneously in infants aged 6 to 7 months with good immunogenicity and safety. As a result, during the influenza season, coadministration of the EV71 vaccine and IIV3 is both feasible and necessary. Since only IIV3 was used in infants aged 6–7 months in China at the time of this study, the immunogenicity and safety of the quadrivalent split-virion influenza vaccine and the EV71 vaccine administered simultaneously need to be confirmed.

## Figures and Tables

**Figure 1 vaccines-11-00862-f001:**
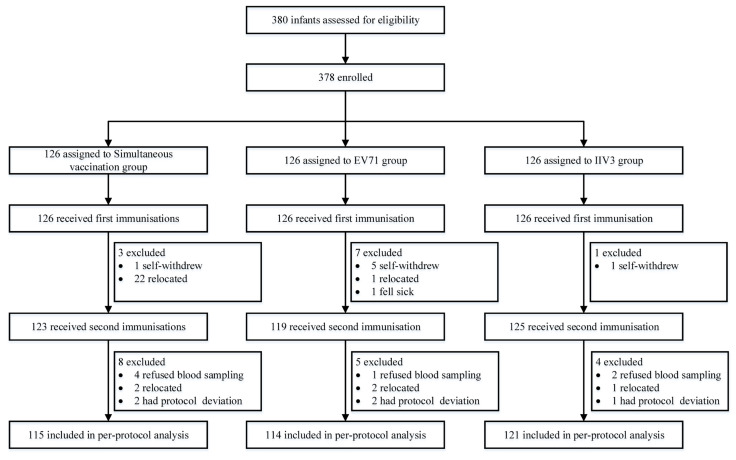
Trial profile. EV71 = inactivated enterovirus 71 vaccine. IIV3 = trivalent split-virion inactivated influenza vaccine.

**Table 1 vaccines-11-00862-t001:** Reported adverse events following any vaccination within 28 days.

	Simultaneous Vaccination Group (*n* = 126)	EV71 Group (*n* = 126)	IIV3 Group (*n* = 126)	*p*
Total, n (%)	31.75	28.57	34.13	0.635
Systemic, n (%)	30.95	27.78	32.54	0.705
Fever	16.67	15.87	12.70	0.648
Rash	5.56	3.97	1.59	0.244
Diarrhea	3.17	1.59	3.97	0.519
Vomiting	0.79	0.79	0.79	1.000
Unsolicited	11.11	11.11	16.67	0.317
Local, n (%)	1.59	0.79	3.17	0.514
Induration	0.79	0.79	2.38	0.626
Unsolicited	0.79	0.00	0.79	1.000

**Table 2 vaccines-11-00862-t002:** Antibody responses to EV71 in the per-protocol population.

	Simultaneous Vaccination Group (*n* = 115)	EV71 Group (*n* = 114)	*p*
Pre-vaccination			
Seropositive Rate (95% CI)	33.04 (24.56~42.43)	35.09 (26.38~44.59)	0.744
GMT (95% CI)	6.04 (5.13~7.10)	6.40 (5.25~7.79)	0.892
Post-vaccination			
Seropositive Rate (95% CI)	100.00 (96.84~100.00)	100.00 (96.82~100.00)	1.000
Seroconversion Rate (95% CI)	98.26 (93.86~99.79)	97.37 (92.50~99.45)	0.643
GMT (95% CI)	414.25 (333.72~514.23)	514.84 (416.17~636.88)	0.157

**Table 3 vaccines-11-00862-t003:** Antibody responses to influenza virus in the per-protocol population.

	Simultaneous Vaccination Group (*n* = 115)	IIV3 Group (*n* = 121)	*p*
H1N1			
Pre-vaccination			
Seropositive Rate (95% CI)	5.22 (1.94~11.01)	3.31 (0.91~8.25)	0.465
GMT (95% CI)	5.31 (5.04~5.60)	5.17 (5.00~5.36)	0.464
Post-vaccination			
Seroprotection Rate (95% CI)	80.87 (72.48~87.61)	86.78 (79.42~92.25)	0.217
Seroconversion Rate (95% CI)	80.00 (71.52~86.88)	86.78 (79.42~92.25)	0.161
GMT (95% CI)	79.52 (64.88~97.46)	88.69 (73.54~106.96)	0.513
H3N2			
Pre-vaccination			
Seropositive Rate (95% CI)	4.35 (1.43~9.85)	9.92 (5.23~16.68)	0.098
GMT (95% CI)	5.18 (5.02~5.36)	5.39 (5.17~5.62)	0.102
Post-vaccination			
Seroprotection Rate (95% CI)	99.13 (95.25~99.98)	98.35 (94.16~99.80)	0.587
Seroconversion Rate (95% CI)	99.13 (95.25~99.98)	98.35 (94.16~99.80)	0.587
GMT (95% CI)	142.69 (125.92~161.69)	125.79 (110.35~ 143.38)	0.133
B			
Pre-vaccination			
Seropositive Rate (95% CI)	1.74 (0.21~6.14)	2.48 (0.51~7.07)	0.692
GMT (95% CI)	5.06 (4.98~5.15)	5.09 (4.99~5.19)	0.694
Post-vaccination			
Seroprotection Rate (95% CI)	76.52 (67.71~83.92)	80.99 (72.86~87.55)	0.401
Seroconversion Rate (95% CI)	76.52 (67.71~83.92)	80.99 (72.86~87.55)	0.401
GMT (95% CI)	49.10 (42.30~56.98)	48.88 (42.37~ 56.39)	0.966

## Data Availability

Data are available for scientific purposes after written request to the corresponding author.

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
