# Peer review of "Safety and Immunogenicity of Enterovirus 71 Vaccine (Vero Cell) Administered Simultaneously with Trivalent Split-Virion Influenza Vaccine in Infants Aged 6–7 Months: A Phase 4, Randomized, Controlled Trial"

_vaccines, 2023, doi:10.3390/vaccines11040862_

Round 1
Reviewer 1 Report
Dear Editor-in-Chief,
I have now read the manuscript entitled: “Safety and immunogenicity of enterovirus71 vaccine (Vero cell) administered simultaneously with trivalent split-virion influenza vaccine in infants aged 6-7 months: A phase 4, randomized, controlled trial” by Xiao Y et al. (manuscript, vaccines-2281136).
Comments: The manuscript describe the use of a combination vaccine developed by Wuhan Institute of Biological Products (the EV71) and by Changcun Institute of Biological Products (the Trivalent-influenza virus vaccines). The vaccines are administered by intramuscular injection to 378 healthy infants at 6-7 months of age. The study and the data look very interesting, but the manuscript lack important information in its current form. More detailed analysis method descriptions are needed to allow fair evaluation of the results.
I here below provide my questions:
Questions:
Q1. How were the vaccines that were used administrated ? (Intramuscularly in the deltoid muscle?).
Q2. In Methods (line 100) the authors describe the EV71 vaccine dose used to be: “containing a neutralizing antibody titer of no less than 3.0EU”?? Was the vaccine against EV71 a passive antibody transfer injection?
Q3. In Methods (line 101-102) no information on which influenza strains were used, this need to be presented. No information is given on how the vaccine was produced (egg, cell-culture, VLP ??). Is it a split or purified HA-subunit vaccine? This information should be presented.
Q4. In Methods (line 100) the EV71-vaccine producer is given but no information is given about the vaccine virus? Was it a formalin-inactivated cell-culture vaccine or was it a adeno-virus recombinant EV71 vaccine strain? Which EV71-virus strain was used for the manufacture of the vaccine?
Q5. In Methods: (Line 114-116). A)How was the cytopathic inhibition assay performed? B) Which virus-strain was used, and on which cells were the inhibition cultures performed? C) How was the inhibition-titer determined? C) Which control serum was used to determine the sensitivity of the neutralization EU-levels?
Q6. Immunogenicity assessment. (line 127-128): The China National Institute for Food and Drug Control provided neutralizing antibodies to EV71 and the influenza viruses. A) Against which strains and subtypes of Influenza A and B viruses were the provided antibodies directed against? B) Which dose of HA was used in the hemagglutination inhibition assay, and which type of erythrocyte was used?
Q7. Randomization and masking: (Line 105) The authors claim that the study was “carried out to decrease needless injections.” How was the needless injections avoided?
Q8. In Table 1 the side effects are presented. Fever seem to be the most frequent side effect, but how high was the fever seen in the children, and for how long duration was the fever lasting?
Q9. In Table 2. The seropositivity against EV71 is presented. It seem like a good amount of children are seropositive already prior to vaccination (24,56 – 44.59%)? What could be the explanation to why 6-7 months young children are EV71 seropositive?
Q10. In Table 3. The influenza virus seropositive response-levels are presented. Here only between 0,91 – 11.01% of the children are seropositive against H1N1 virus at 6-7 months of age, before vaccination. What could be the reason for this levels of seropositive reactivity in these young children before vaccination? At what time of the year (winter, summer, spring?) was the study performed?
Q11. In Table 3 the serum-viral neutralization GMT:s are shown, but since there is no decription of the levels of virus was used in the virus-inhibition assays it is difficult to interpret if the obtained “protective titers” are relevant or not? It too low titers of virus (EV71) or hemagglutinin (InfluenzaA and B) was used then the results may show a too optimistic level of “protective humoral immunity”?
Reviewer 2 Report
The manuscript describes the results of a phase 4 of the safety and immunogenicity of the simultaneously administration of enterovirus 71 vaccine (EV71 vaccine) with the trivalent split-virion influenza vaccine (IIV3) in infants aged 7-8 months. The results show that the simultaneously administration does not interfere with the safety of each vaccine in particularly and also with the potential to induce protective antibody production in the infants.
Despite that, some points should be better discussed and altered to improve the impact of the review. The authors should add more information about the phases 1, 2 and 3 of the vaccines used in the study. Furthermore, the main reason for choosing the age of 6-7 months for the infants in the study considering the incidence of these virus infections.
Methods section:
The authors should describe in detail the cytopathic inhibition method for the EV71 vaccine that support the results of seroconversion. In addition, the references that support the methodology should be added.
For the influenza virus antibody detection, the reference that support the assay and the details of the methodology should be also added.
Discussion section.
The authors concluded that the co-administration of EV71 and IIV3 in infants is safe and induce a robust antibody response and, therefore can be protective against these virus infections. In this sense, the authors should include some information comparing the levels of the antibody production obtained with others studies of different vaccines in infants with same age and/or studies with these vaccines in adults to improve the discussion.
Reviewer 3 Report
In this article the authors presented the good results obtained after simultaneously immunization of the infant aged 6-7 months with EV 71 vaccine and IIV3 .
Author Response
Thank you for your comment.
Round 2
Reviewer 2 Report
All suggested changes are included in the resubmitted version by the authors, therefore, the article is now improved for publication.